# Article

# Eigendegradation Algorithm Applied to Visco-Plastic Weak Layers

Pedro Navas [1], Diego Manzanal [1,*], Ángel Yagüe [1], Miguel M. Stickle [1] and Susana López-Querol [2]

1   ETSI Caminos, Canales y Puertos, Universidad Politécnica de Madrid, Prof. Aranguren 3,
    28040 Madrid, Spain
2   Department of Civil, Environmental and Geomatic Engineering, University College London, Gower Street,
    London WC1E 6BT, UK
*   Correspondence: d.manzanal@upm.es

**Abstract:** In geotechnical engineering, very often, the soil behavior varies with time. This is of particular interest in many cases such as embankments in soft clays, shear band progression in slopes or where the speed of the application of the load affects the bearing capacity of the material. In this paper, we study the extension of non-local failures using algorithms such as eigenerosion and eigensoftening, in order to evaluate the failure of weak layers. In particular, the time dependence of the progression of shear bands is analyzed through the integration of a Perzyna-type visco-plastic model with a degradation algorithm within the Optimal Transportation Meshfree (OTM) framework. The validation of the proposed algorithm is carried out through three different practical cases, showing very good agreement in all of them.

**Keywords:** meshfree numerical modeling; finite deformation; degradation; sensitive clays; visco-plastic behavior





## 1. Introduction

The failure of geomaterials has been a topic of study since the early work of Coulomb back in 1877. The work of Drucker and Prager [1] is an example of the theoretical derivation of these problems. Failures can be classified in two types depending on the state of the material: diffuse and localized.

Diffuse failures are associated with the instability of loose and saturated materials (e.g., [2,3]) under monotonic and dynamic loads. In the latter, the increase of the pore pressure can cause the liquefaction of the material, when its effective stress vanishes as a result of this increase of pore pressure while the total stress remains constant. This phenomenon can produce landslides, involving the displacement of large volumes of material. The numerical modeling of the initiation of diffuse failure has been developed in finite element based codes to reproduce failures in mine tailing dams [4,5], landslides [6,7] or the study of the response of loose saturated, granular material under seismic action [8,9]. Recently, these phenomena have been studied with meshfree techniques, allowing to evaluate the transition from solid to fluidized materials with SPH techniques with one (e.g., [3,10–14]) and two phases [15,16], Material Point Method, MPM, (e.g., [17–20]) or Discrete Element Methods, DEM, for the liquefaction on granular materials (e.g., [21,22]).

In contrast, localized failures are associated with the concentration of strains in a narrow zone and a limited region, which produces a discontinuity in the deformation as well as in the strain rate. This failure is related to a weak discontinuity, which is also known as a weak layer. The progress of the failure of overconsolidated clays that present a softening behavior has been extensively studied. It consists of the reduction of the shear strength from the peak to the residual value. Early work by Rice [23] provided the theoretical basis for understanding the physics and mathematics of the problem. The formation and propagation of shear bands have been studied through theoretical and lab-scale models

(e.g., [24,25]). Numerically, the localization of shear deformations associated with softening behaviors, which is related to negative slopes in the loading–displacement curve, traditionally depends on the mesh discretization (e.g., [26]. The employment of visco-plastic models has allowed the regularization of the problem (e.g., [27]).

Visco-plastic formulations are common in the literature to evaluate the rate-dependent failure of geomaterials, shear bands, creep and stress relaxation. In general, such constitutive formulations include empirical models focused on the evaluation of creep and stress relaxation for soft clays [28,29], rheological models for a wide range of deformations (e.g., [11,30,31]) and general stress–strain–time constitutive models (e.g., [32,33]). Elasto-plastic constitutive models incorporating rate dependence are based mainly on the over-stress theory [34] and on the concept of a non-stationary flow surface (e.g., [35,36]). The over-stress theory assumes there is no viscous strains occurring inside the static yield surface (elastic region). However, the inelastic region is rate dependent. This dependency is defined by an over-stress factor. This theory has been implemented in finite element and meshfree methods to study different geotechnical problems (e.g., [31,37]).

In this paper, we propose a methodology for evaluating the progression of a shear band or a weak layer through a phenomenological visco-plastic model, taking the benefits of a non-local failure approach. The proposed algorithm is based on the same principles of both eigenerosion and eigensoftening approaches, which have been widely employed through material point-based frameworks in order to assess the fracture evolution in quasi-brittle materials ([38,39]) once specific energy (eigenerosion) or stress (eigensoftening) thresholds are reached when the material point fails. Within the aforementioned algorithms, the failure can be modeled instantaneously (eigenerosion) or following a material-dependent softening law (eigensoftening). In order to achieve a diffuse failure through the degradation of the material, a combination of these algorithms within a Perzyna-type visco-plastic model is proposed. This methodology is validated together with a particle-based discretization (OTM) with three practical cases: (a) the progression of a shear band, (b) the effect of the velocity on the ultimate load of a footing and (c) the failure of a vertical slope with a weak layer.

The Optimal Transportation Meshfree [40–42] is a meshfree method that has been demonstrated to perform reasonably well in geotechnical problems [43,44]. Its discretization is based on an FEM scheme, where the nodes map the displacement field (and its derivatives) and the material points (originally located at the integration points) carry the material information such as energy, stress and strain. In contrast to the FEM, the information of the material point comes from a neighborhood of nodes, which is defined by the distance between them and the node, instead of a predefined element. The main advantage of this is that the neighborhood is changing every step, which avoids the element distortion, since the new neighborhood will be adapted to the new shape of the domain.

Therefore, with the aim of verifying the good performance of the aforementioned algorithm, the rest of the paper is organized as follows; in Section 2, the constitutive model with all its ingredients (rate-dependent plasticity and non-local failure procedure) is presented; in Section 3, time and spatial discretizations are outlined; in Section 4, several applications are depicted; and, finally, the derived conclusions are provided in Section 5.

## 2. Constitutive Model

### 2.1. Rate Dependent Plasticity

Apart from the mathematical and numerical models that will be described in Section 3, it is essential to choose an appropriate constitutive model for the degradable soils. The mechanisms through which soft clays display a viscous phenomenon and delayed deformation because of the creep have been extensively studied in the past ([28]). The concept of the visco-plastic material outlined in this section and employed in the proposed simulations is based on Perzyna's theory [34], which is a modification of the classical plasticity, wherein viscous-like behavior is introduced by a time-rate flow rule thanks to a plastic yield function adapted to dynamic conditions. In a similar manner to the rate-independent theory, the strain rate is split into a visco-plastic strain rate and an elastic one:

$$\dot{\varepsilon} = \dot{\varepsilon}^{\mathrm{e}} + \dot{\varepsilon}^{\mathrm{vp}} \tag{1}$$

The rate of the Cauchy's stress tensor, $\dot{\sigma}$, is linked to the elastic strain rate through the constitutive tensor $\mathbf{D}_e$, which, in our case, is variable or stress-dependent due to the hyperelasticity and large strain theory.

$$\dot{\sigma} = \mathbf{D}_e(\dot{\varepsilon} - \dot{\varepsilon}^{vp}) \tag{2}$$

In the model proposed by Perzyna [34], and later modified by Souza-Neto et al. [45], the rate of the visco-plastic strain can be defined similarly than in the rate-independent plasticity approach:

$$\dot{\varepsilon}^{\mathrm{vp}} = \langle \dot{\lambda} \rangle \frac{\partial g}{\partial \sigma} \tag{3}$$

where $\langle \dot{\lambda} \rangle$ is the function of the viscous flow, which denotes the current magnitude of visco-plastic strain rate; $g$ represents the visco-plastic potential function and $\frac{\partial g}{\partial \sigma}$ represents the current direction of the visco-plastic strain rate. The viscous flow function is defined by:

$$\langle \dot{\lambda} \rangle = \begin{cases} \gamma \left[ \left( \frac{q}{\sigma_y} \right)^{\alpha} - 1 \right] & , \phi > 0 \\ 0 & , \phi \le 0 \end{cases}, \tag{4}$$

where $<>$ denotes Macauley brackets, $\gamma$ is the fluidity parameter, also thought as the reciprocal of the viscosity, and $\alpha$ is a material constant. In this work, associative flow is invoked by $\phi = g$. $\phi$ is the plasticity function that plays the role of the loading surface; a von Mises $\phi = g$ function, with a degradation curve for the undrained shear strength, has been adopted for the plastic criteria. Function $\phi$ is defined as:

$$\phi = q - \sigma_y, \tag{5}$$

where $q$ is the deviatoric stress invariant and $\sigma_y$ is defined by the degradation law, which will be detailed later. Regarding algorithmic aspects, in displacement-based numerical methods, stress updates take place at the Gauss points (or material points if material-based methodologies are adopted) for a known nodal displacement. Initial conditions, which are the ones from the last converged state, depart from time $t_n$:

$$\left[ \varepsilon_n, \varepsilon_n^{vp}, \sigma_n, \kappa_n \right], \tag{6}$$

in which the variables are total strain, visco-plastic strain, stress and a scalar internal variable, respectively. The latter characterizes the size of the yield surface as well as any other plastic aspect. The final goal is to calculate the corresponding values at time $n + 1$ through:

$$t_{n+1} = t_n + \Delta t : \left[ \varepsilon_{n+1}, \varepsilon_{n+1}^{vp}, \sigma_{n+1}, \kappa_{n+1} \right]. \tag{7}$$

Indeed, this process has been carried out in an incremental way, being calculated:

$$\Delta \varepsilon = \varepsilon^{\mathrm{e}} + \Delta \varepsilon^{\mathrm{vp}} \tag{8}$$

$$\Delta \sigma = \mathbf{D}_{\mathrm{e}}(\Delta \varepsilon - \Delta \varepsilon^{\mathrm{vp}}) \tag{9}$$

Thus, the main objective of the stress updates is to estimate the visco-platic strain increment, $\Delta \varepsilon^{vp}$. The numerical implementation of this calculation follows the textbooks of Owen and Hinton, 1986 [46] and De Souza Neto et al., 2008 [45]). It has to be pointed out that the visco-plastic approach has a regularizing effect when softening behavior is to be modeled (Wang et al. [26]) since the initial-value problem remains well-posed, avoiding instability due to the suffered softening of strain and strain-rate.

### 2.2. Eigenerosion and Eigensoftening Algorithms

Within the context of the OTM formulation, fracture can be modeled simply by eroding material points according to an energy-release or stress criteria, depending on whether the eigenerosion [47–50] or the eigensoftening [38,39,51] algorithm are adopted, respectively. In both methodologies, if the material points failed, in order to approximate the presence of cracks, the material points can be extracted from the computation of stresses. However, the way to reach the zero stiffness state is different for each model: meanwhile, in the eigenerosion, the failure is instantaneous, in the eigensoftening, the material follows a softening curve, which depends on the material. It needs to be noticed that when a material point satisfies the failure condition, its contribution to the material stiffness matrix, as well as to the internal forces, is set to zero. However, it is necessary to maintain its contribution to the mass matrix in order to fulfill the mass conservation law. We only can discard it if an eroded material point is finally unconnected to any other node.

In the following, we compute the energy-release rate attendant to the failure of the material point $p$, which is the starting point of the aforementioned methodologies:

$$
\begin{aligned}
G_{p,k+1} &= \frac{C\epsilon}{m_{p,k+1}} \sum_{x_{q,k+1} \in B_\epsilon(x_{p,k+1})} m_q W_k(F_{q,k+1}), \\
m_{p,k+1} &= \sum_{x_{q,k+1} \in B_\epsilon(x_{p,k+1})} m_q,
\end{aligned}
\tag{10}
$$

where $W_k(F_{q,k+1})$ is the free-energy density per unit mass at the material point $x_{q,k+1}$, and $B_\epsilon(x_{p,k+1})$ is the $\epsilon$-neighborhood of the material point, which is calculated as the volume of $\epsilon$ size centered at the aforementioned $x_{p,k+1}$. $m_{p,k+1}$ is the mass of this neighborhood and $C$ is a normalizing constant, which ranges from 1 to 2, in order to extend the influence from the first line of neighbors to the second [47,48]. Everything is evaluated at the time step $k+1$. A scheme of the $\epsilon$-neighborhood and how it is configured is plotted in Figure 1.

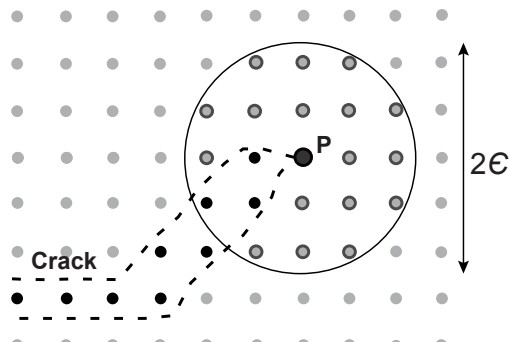

**Figure 1.** Scheme of a set of fractured material points as a fractured layer (black dots) and the $\epsilon$-neighborhood of the points neighbors of the material point at the crack tip (gray dots).

The material point is failed when the value of $G_{p,k+1}$ (Equation (10)) exceeds the critical energy release rate, $G_F$. This parameter estimates the specific energy required to create a fracture surface per unit of area. In the proposed algorithm, every time step the eroded (failed) material-point set was updated, taking into account this criterion. Schmidt et al. [47] have demonstrated that this approximation converges to Griffith fracture for linear elasticity when an infinitely fine discretization is presumed. Certainly, schemes that estimate the energy-release rate based on the energy of a single material point may suffer from an overestimation of the toughness of the material as well as mesh-dependency if non-local approaches are adopted.

On the other hand, the implementation of the eigensoftening algorithm consists of adopting a strength criterion for crack initiation and a law which is capable of reproducing a proper reduction of strength of the material under study before the formation of a stress-free

crack, which is known as softening. This second process tends to accumulate less energy until the crack appears. When the tensile strength, $f_t$, is achieved, a crack is conformed with zero opening displacement. Because of its cohesive behavior, once the opening displacement, $w$, reaches a critical vale, $w_c$, a stress-free crack is attained. The energy below the softening curve represents the fracture energy per unit of area in static conditions, $G_F$, which is sketched in Figure 2 [38].

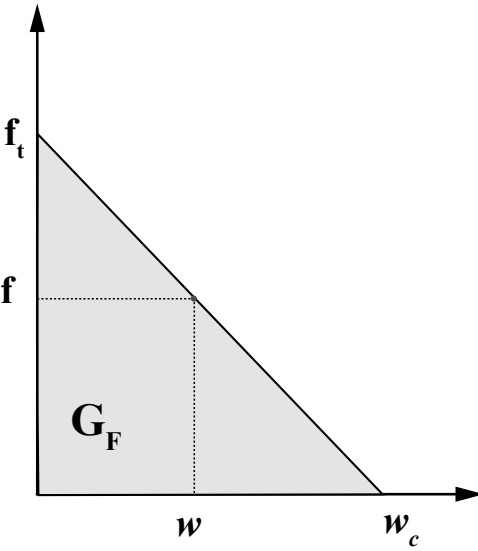

**Figure 2.** Scheme of a cohesive law, where $f_t$ is the tensile strength, $w_c$ is the critical opening displacement and the shade area, calculated through $f_t$ and $w_c$, is $G_F$.

For the eigensoftening calculation, Equation (10) can be rewritten in terms of the principal stresses at time $t_{k+1}$, since this model employs the first principal stress as a failure criterion. Therefore, the increment of the averaged density of the strain energy in a $\epsilon$-neighborhood of the material point $x_{p,k+1}$ may be calculated as

$$\delta W_p^\epsilon = \frac{\partial G_p}{C\epsilon} = \frac{1}{m_p} \sum_{x_q \in B_\epsilon(x_p)} m_q \sigma_{q,1} \delta \varepsilon_q, \tag{11}$$

in which $\sigma_{q,1}$ is the maximum tensile stress (principal stress 1) at a material point $x_{q,k+1}$ from the neighborhood. Considering, for a material point $x_{q,k+1}$, an effective strain $\varepsilon_q$ such that the variation of the local strain energy can be calculated as $\delta W_q = \sigma_{q,1} \delta \varepsilon_q$, let us assume the effective strain increment can be approximated, for each material point, by its counterpart in the neighborhood, in which Equation (11) is simplified as:

$$\delta W_p^\epsilon = \frac{\delta \varepsilon_p}{m_p} \sum_{x_{q,k+1} \in B_\epsilon(x_{p,k+1})} m_q \sigma_{q,1}. \tag{12}$$

Thus, the equivalent critical stress at the material point $x_{p,k+1}$ is defined as follows

$$\sigma_p^\epsilon = \frac{1}{m_p} \sum_{x_{q,k+1} \in B_\epsilon(x_{p,k+1})} m_q \sigma_{q,1} \tag{13}$$

When $\sigma_{p,k+1}^\epsilon$ surpasses the tensile strength, $f_t$, the softening behavior is activated through the damage variable $\chi$, which ranges between zero (an intact material) and one (completely failed material points). Of course, $\chi$ depends on the current and critical opening measures, $w$ and $w_c$, respectively. The latter is a material parameter, but the first one has to be measured in terms of the achieved strain and a length of affection called bandwidth, $h_\epsilon$, which is equivalent to the crack band model of Bažant [52]. It must be pointed out that, according

to Bažant [53], the reference value for $h^\epsilon$ ranges between two and four times the maximum size of the aggregates for concrete. Thus, this is a material parameter more than a numerical artifact. The relationship between strain and crack opening depends on the effective fracture strain, $\varepsilon^\epsilon_f$, which is defined as the difference between the strain at crack initiation, $\varepsilon_1(x_{p,0})$, and the current strain, $\varepsilon_1(x_{p,k+1})$ for a material point $p$; and the bandwidth is:

$$\varepsilon^\epsilon_f = \varepsilon_1(x_{p,k+1}) - \varepsilon_1(x_{p,0}) = \frac{w}{h^\epsilon} \tag{14}$$

### 2.3. Eigendegradation Model

Following the work of Einav and Randolph [54], and the later implementations by Zhang et al. [54] (similar also to some other implementations [55,56]), the behavior of sensitive clays can be modeled by strain-softening curves in order to reduce the strength of the material by a degradation related to the accumulation of strain. Einav and Randolph assumed that the current shear strength depends on the accumulated absolute shear strain, $\xi$, which is assumed as a state variable in order to calculate the isotropic strength reduction, $\delta(\xi)$, as:

$$\delta(\xi) = s_u / s_{ui} = \delta_{\text{rem}} + (1 - \delta_{\text{rem}})e^{-3\xi/\xi_{95}}, \tag{15}$$

where

$$\xi = \int_t |\dot{\gamma}_{\text{max}}| \mathrm{d}t \tag{16}$$

and $|\dot{\gamma}_{\text{max}}|$ represents the cumulative absolute shear strain. $s_u$ and $s_{ui}$ are considered the softened strength and initial strength, respectively. In the above equations, $\delta_{\text{rem}}$ is the fully remolded strength ratio, and $\xi_{95}$ is the cumulative shear strain required to cause 95% reduction from the peak to fully remolded material. An appropriate value of $\xi_{95}$ must be obtained from laboratory test data as well as from cyclic penetration and extraction tests with T-bar or ball. Furthermore, $\delta_{\text{rem}}$ is assumed to be the inverse of the sensitivity of the soil.

The calculation of the cumulative shear strain can be achieved by the eigendeformation technique, departing from Equation (11), and considering that, for the calculation of the eigendegradation, that the stress remains constant in a neighborhood $\epsilon$. Thus, Equation (11) can be simplified as

$$\delta W^\epsilon_p = \frac{\delta \tau_p}{m_p} \sum_{x_{q,k+1} \in B_\epsilon(x_{p,k+1})} m_q \gamma_q, \tag{17}$$

where $\delta \tau_p$ refers to the increment of tangential stress of the neighborhood and $\gamma^\epsilon_p$ is the current local shear strain, which is obtained as:

$$\gamma^\epsilon_p = \frac{1}{m_p} \sum_{x_{q,k+1} \in B_\epsilon(x_{p,k+1})} m_q \gamma_q \tag{18}$$

Similarly, in the neighborhood $\epsilon$, the non-local cumulative strain of a material point $p$ is calculated only when plasticity is activated, as follows:

$$\xi^\epsilon_p = \int_{t_{p0}}^{t_{k+1}} \left| \dot{\gamma}^\epsilon_p \right| \mathrm{d}t \tag{19}$$

where $t_{k+1}$ refers to the current step, and $t_{p0}$ refers to the step when plasticity begins.

Considering only shear failure, yield shear stress $\tau$ is equivalent to the softened strength, $s_u$, and the residual yield shear stress, $\tau_{95}$, can be reached by $\tau_{95} = \tau rem = s_{ui} \delta_{rem}$. Thus, in every state of degradation, the current yield shear stress, referring to the epsilon neighborhood, $\tau^\epsilon$, reads:

$$\tau^\epsilon = \tau_{95} + (\tau_i - \tau_{95})e^{-3\xi^\epsilon_p/\xi_{95}} \tag{20}$$

It is remarkable that in the laboratory, the parameter $\zeta_{95}$ is not obtained. Instead, the displacement $\delta_{95}$ is achieved. In Figure 3A, the degradation of the strength in terms of the displacement is plotted.

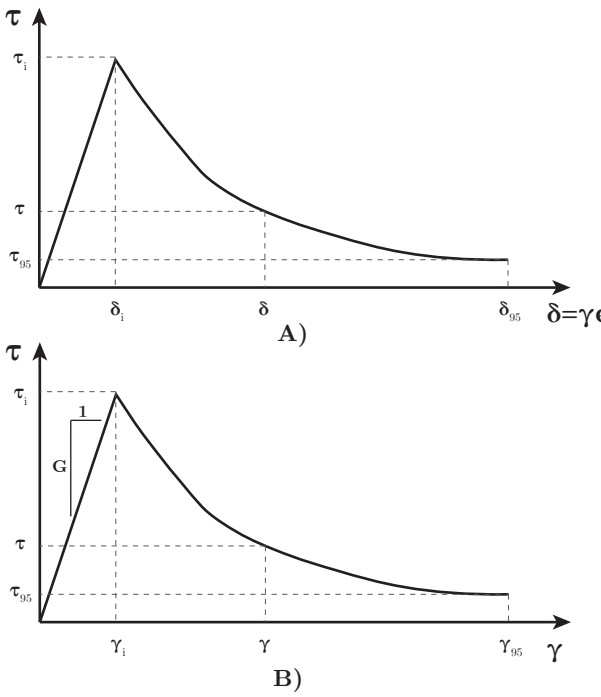

**Figure 3.** Degradation curve in terms of the displacement (**A**) and the shear strain (**B**).

It can be seen how this law can be translated to the shear strain measurement (Figure 3B) by multiplying by $\epsilon$, which, in this problem, is considered as the sliding length. Depending on the size of the soft layer, this parameter $\epsilon$ is obtained as the minimum length between the neighbor radius, [38], and the size of the soft layer (see Figure 4) as follows:

$$2\epsilon = \min(h_s, 2C_\epsilon h) \tag{21}$$

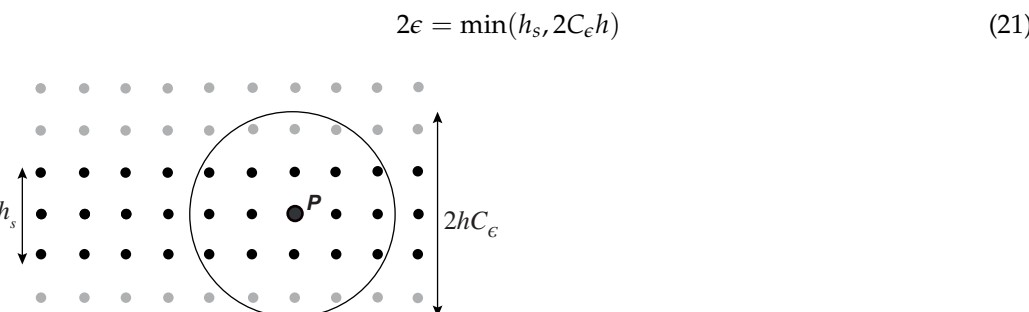

**Figure 4.** Scheme of the measurements of the soft layer (black dots) and the $\epsilon$-neighborhood around material point P.

### 2.4. Visco-Plastic Eigendegradation Algorithm

Following, the pseudo-algorithm for the eigendegradation model within a visco-plastic yield surface will be presented. It is worth mentioning that prior to the algorithm steps, we need to calculate the equivalent shear total strain of every material point as the norm of the deviatoric total strain tensor. Since large strain is considered, the strain tensor is obtained through the logarithm of the left Cauchy–Green strain tensor, **b**:

$$\varepsilon = \frac{1}{2}\log \mathbf{b} = \frac{1}{2}\log \mathbf{FF}^T. \tag{22}$$

The aforementioned procedure can be followed in Algorithm 1.

---

**Algorithm 1** Visco-Plastic Eigendegradation algorithm.

---

**1. Calculation of the small strain tensor**

$$\varepsilon_{k+1}^{e\ trial} = 1/2 \log \mathbf{b}_{k+1}^{e\ trial}$$

**2. Elastic predictor: volumetric and deviatoric stress measurements**

$$
\begin{aligned}
\text{Volumetric:} &\quad p_{k+1}^{trial} = K\left(\varepsilon_{vol}^{e}\right)_{k+1}^{trial} \\
\text{Deviatoric:} &\quad \mathbf{s}_{k+1}^{trial} = 2G\left(\varepsilon_{dev}^{e}\right)_{k+1}^{trial} \\
\text{being:} &\quad \sigma_{k+1}^{trial} = J^{-1}\tau_{k+1}^{trial} \\
\text{and:} &\quad q_{k+1}^{trial} = \sqrt{\tfrac{3}{2}} \|\mathbf{s}_{k+1}^{trial}\|
\end{aligned}
$$

**3. Eigendegradation calculation**:

**if** $t < t_{p0}$ **then** $\quad \sigma_y = \tau_i$
**else**

- $m_p = \sum_{x_{q,k+1} \in B_\epsilon(x_{p,k+1})} m_q$
- $\gamma_p^\epsilon = \frac{1}{m_p} \sum_{x_{q,k+1} \in B_\epsilon(x_{p,k+1})} m_q \gamma_q$
- $\xi_p^\epsilon = \sum_{k(t_{p0})}^{k+1} \left|\Delta\gamma^\epsilon(x_{p,k})\right|$
- $\sigma_y = \tau_{95} + (\tau_i - \tau_{95})\mathrm{e}^{-3\xi_p^\epsilon/\xi_{95}}$
- Hardening modulus: $\quad H = \frac{\partial \sigma_y}{\partial \bar{\varepsilon}^p} \simeq \frac{\partial \sigma_y}{\partial \xi_p^\epsilon} = -\frac{3(\tau_i - \tau_{95})}{\xi_{95}}\mathrm{e}^{-3\xi_p^\epsilon/\xi_{95}}$

**end if**

**4. Yield condition**: $\Delta\lambda = 0$

**if** $\phi = q_{k+1}^{trial} - \sigma_y \leq 0$ Elastic region: $\quad \sigma_{k+1} = \sigma_{k+1}^{trial}$
**else** $\quad$ Visco-plastic flow:

- **4.1** Derivative of the yield surface:

$$d = \frac{\partial \phi}{\partial \Delta\lambda} = -\left(3G + \alpha\frac{q_{k+1}^{trial} - 3G\Delta\lambda}{\Delta\lambda + \gamma\Delta t}\right)\left[\frac{\gamma\Delta t}{\Delta\lambda + \gamma\Delta t}\right]^\alpha - H$$

- **4.2** Increment of plastic strain: $\Delta\lambda = \Delta\lambda - \frac{\phi}{d}$

- **4.3** Yield function: $\quad \phi = \left(q_{k+1}^{trial} - 3G\Delta\lambda\right)\left[\frac{\gamma\Delta t}{\Delta\lambda + \gamma\Delta t}\right]^\alpha$
- **4.4** If $\phi < tolerance$ go to **4.5**, else go to **4.1**
- **4.5** Update

$$\bar{\varepsilon}_{k+1}^p = \bar{\varepsilon}_k^p + \Delta\gamma$$

$$\Delta\varepsilon_{k+1}^p = \frac{\Delta\gamma}{\|\mathbf{s}_{k+1}^{trial}\|}\mathbf{s}_{k+1}^{trial}$$

$$\sigma_{k+1} = \left(p_{k+1}^{trial}\right)\mathbf{I} + \left(1 - \frac{3G\Delta\gamma}{q_{k+1}^{trial}}\right)\mathbf{s}_{k+1}^{trial}$$

**end if**

**5. Update elastic left Cauchy–Green Tensor**

$$\varepsilon_{k+1}^e = \varepsilon_{k+1}^{e\ trial} - \Delta\varepsilon_{k+1}^p$$

$$\mathbf{b}_{k+1}^e = \exp(2\varepsilon_{k+1}^e)$$

### 3. Time and Spatial Discretization

Following, the rest of the computational tools that were employed in the present research are highlighted. First, the spatial discretization is shown, while the time discretization is mentioned in the final subsection.

### 3.1. Spatial Discretization

The dynamic problem of a dry soil (mono-phase material) is studied in this research, in which the time is an important issue in the following analyses. The formulation of the dynamic problem can be defined by the governing equation of the balance of the linear momentum:

$$\text{div } \sigma - \rho a + \rho g = \mathbf{0}. \tag{23}$$

The weak form derivation, following the Galerkin procedure, needs to multiply (23) by the virtual displacement (a test function) $\delta u$, and the integration over the domain. After applying Green's theorem, Equation (23) reads as follows:

$$-\int_{\Omega} \sigma : \text{grad}\delta u \, \mathrm{d}\Omega + \int_{\Gamma_t} \delta u \cdot \bar{t} \, \mathrm{d}\gamma - \int_{\Omega} \delta u \cdot \rho a \, \mathrm{d}\Omega + \int_{\Omega} \delta u \cdot \rho g \, \mathrm{d}\Omega = 0 \tag{24}$$

where $\Omega$ is the volume of the body, and the boundary where tractions are applied is depicted by $\Gamma$. The internal forces of the body are represented by the first term of the equation; meanwhile, the external forces are represented by the second and fourth terms. The third one refers to the inertial terms. The next step is the interpolation through the Optimal Transportation Meshfree. As we mentioned, this numerical method is based on a calculation framework of nodes and material points. The shape functions are based on the work of Arroyo and Ortiz [57], where the Local Max-Ent shape function (LME) of the material point $(x)$ with respect to the neighborhood $(x_a)$ is defined as follows:

$$N_a(\mathbf{x}) = \frac{\exp\left[-\beta_{LME} |\mathbf{x} - \mathbf{x_a}|^2 + \lambda^* \cdot (\mathbf{x} - \mathbf{x_a})\right]}{Z(\mathbf{x}, \lambda^*(\mathbf{x}))}, \tag{25}$$

in which $Z(\mathbf{x}, \lambda)$ is computed along a neighborhood $N_b$ as:

$$Z(\mathbf{x}, \lambda) = \sum_{a=1}^{Nb} \exp\left[-\beta_{LME} |\mathbf{x} - \mathbf{x_a}|^2 + \lambda \cdot (\mathbf{x} - \mathbf{x_a})\right]. \tag{26}$$

The derivatives of this shape function can be obtained from the Hessian matrix $\mathbf{J}$ as follows:

$$\nabla N_a^* = -N_a^* (\mathbf{J}^*)^{-1} (\mathbf{x} - \mathbf{x_a}), \tag{27}$$

The value of parameter $\beta_{LME}$ is associated to the shape of the neighborhood as well as the discretization size (or nodal spacing), $h$. Both parameters are related through parameter $\gamma_{LME}$, which controls the locality of the shape functions, as it is observed following:

$$\beta = \frac{\gamma_{LME}}{h^2}. \tag{28}$$

It bears pointing out that $\lambda^*(\mathbf{x})$ is obtained through the minimization of the function $g(\lambda) = \log Z(\mathbf{x}, \lambda)$ to guarantee the maximum entropy.

By employing the outlined shape functions and applying Galerkin procedure to the weak form, $u$ can be interpolated by employing:

$$u \approx u^h = \mathbf{N}_u \cdot \tilde{u} \tag{29}$$

where $\square^h$ represents the OTM approximation of the field $\square$ and $\tilde{\square}$ represents the nodal values. $N_u = [N_1 I, N_2 I, ..., N_m I]$ is the shape function, in which $m$ is the number of neighbor nodes. The shape functions are defined in the updated configuration, $N = N(x)$. Moreover, the following property helps to calculate time variations: $\dot{\square}^h = N \cdot \dot{\tilde{\square}}$.

*3.2. Time Discretization*

In this work, an implicit scheme has been proposed, since several applications cover a wide range of loading rates from slow scenarios to quick phenomena. For the first ones, an explicit scheme would provide long computation time. Thus, the Newmark Implicit Scheme has been employed, with the parameters $\gamma = 0.6$ and $\beta = 0.325$ that are known to be suitable for dynamic problems [58]. To construct this scheme, Equation (24) is reformulated as a system of equations, which reads as

$$R_{k+1} + M\, \ddot{u}_{k+1} = P_{k+1}, \tag{30}$$

where $R$ and $M$ denote the internal forces vector and mass matrix, respectively. $P$ holds for the external forces vector, which is composed by both gravity acceleration and external nodal forces as we have seen previously. Equation (30) can be re-written with the Newmark scheme as:

$$G_{k+1} \quad = \quad M[\alpha_1 \Delta u_{k+1} - \alpha_2 \dot{u}_k - \alpha_3 \ddot{u}_k] + R_{k+1} - P_{k+1} = 0, \tag{31}$$

where the $\alpha$-parameters are calculated according to Wriggers [59] and are listed in Table 1. These coefficients can be easily extended to any other time integration schemes.

**Table 1.** The $\alpha$-parameters of the Newmark scheme.

| $\alpha_1 = \frac{1}{\beta \Delta t^2}$ | $\alpha_2 = \frac{1}{\beta \Delta t}$ | $\alpha_3 = \frac{1}{2\beta} - 1$ |
| --- | --- | --- |

When the above non-linear equations are solved through a Newton–Raphson method, the resulting iterative scheme, taking into account the matrices that are involved in our problem, can be written as:

$$\left[\alpha_1 M + K^i_{k+1}\right] \Delta u^{i+1}_{k+1} = [K_*]^i_{k+1} \Delta u^{i+1}_{k+1} \quad = \quad -G(u^i_{k+1}), \tag{32}$$

$$\text{where} \quad u^{i+1}_{k+1} \quad = \quad u^i_{k+1} + \Delta u^{i+1}_{k+1}.$$

where $K$ is the derivative of the internal forces of each iteration $i$, which is also known as the *tangential stiffness matrix*:

$$K(u^i_{k+1}) = K^i_{k+1} = \left. \frac{\partial R}{\partial u} \right|_{u^i_{k+1}}. \tag{33}$$

The iteration procedure finishes when the norm of the residuum $G^i_{k+1}$ is below a given tolerance.

## 4. Applications

The previously described methodology has been applied to three examples in this paper. The first two of them are devoted to show the performance of the two main principal properties of the proposed constitutive model: the degradation (Shear test, Section 4.1) and the viscous behavior (Strip footing load, Section 4.2). The last example shows the suitability of the model when the triggering and propagation of a slope due to cyclic loading is sought.

### 4.1. Shear Test

In the first example, the behavior of a weak layer under a shear test is analyzed. In Figure 5, an embankment of 10 m of depth is presented. A weak layer in the bottom of the domain of 0.5 m appears. The proposed shear test is similar to the one proposed by Zhang et al. [54]. Although different constitutive laws are employed (local and non-local), it is expected to obtain comparable results since both laws are based on similar degradation processes.

The original embankment's length is 700 m. Since the softened zone only extends 90 m and considering infinite conditions at both sides of this softened zone, only this 90 m is modeled in order to save computational effort. Unlike the original example, gravity conditions are neglected, considering the failure of the embankment at the final of the residual strength. Thus, the parameters needed in this example are shown in Table 2. It is important to point out that in this research, the proposed non-local degradation model has been employed with a neighborhood parameter, $C_\epsilon$, of 1.5. This is an important difference with respect to the model proposed for the original example, which evaluates the degradation locally in an Arbitrary Eulerian–Lagrangian (ALE) configuration.

**Table 2.** Parameters for the shear degradation analysis.

| | |
|---|---|
| Softened/modeled length $L = l_0$ | 90 m |
| Overall height, $H$ | 10 m |
| Height of sliding material, $h$ | 7.2 m |
| Shear band thickness, $s$ | 0.5 m |
| Submerged density of the soil, $\rho$ | 600 kg/m$^3$ |
| Poisson's ratio, $\nu$ | 0.495 |
| Young's modulus, $E$ | 1.98 MPa |
| Peak shear strength, $\tau_p = \tau_i$ | 10 kPa |
| Residual (95%) shear strength, $\tau_{95} = \tau_r$ | 1.25 kPa |
| Plastic shear strain to 95% reduction in strength, $\gamma_p$ | 0.6 |
| Neighborhood parameter, $C_\epsilon$ | 1.5 |

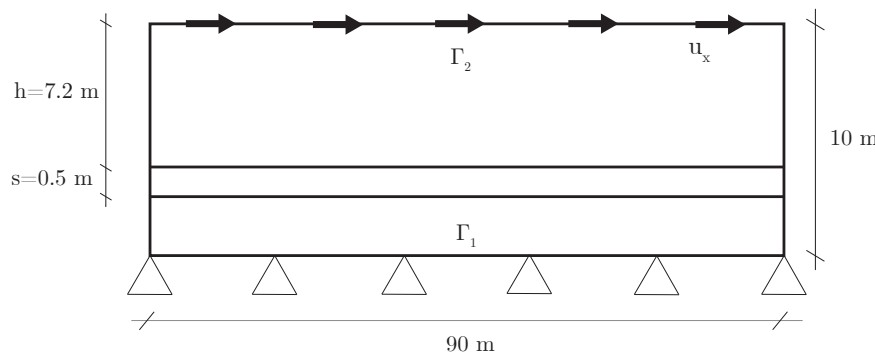

**Figure 5.** Geometry and loading conditions of the shear degradation problem.

In addition, the geometry and boundary conditions can be seen in Figure 5. Regarding the latest, two boundary conditions have been considered. In the first one, $\Gamma_1$ in Figure 5, both vertical and horizontal displacements are constrained. In $\Gamma_2$, a horizontal displacement of 10 m is imposed gradually from 0 to 1000 s.

The stress behavior is analyzed in Figure 6. In order to assess the performance of the proposed algorithm, similar to the figure proposed by Zhang et al. [54], in Figure 6, a dimensionless measurement of the shear stress is plotted. It is calculated as a relative increment from the residual shear strength $\tau_{95}$ and divided by the maximum increment, which is measured from the initial (or peak) strength to the residual one, $\tau_p - \tau_{95}$. On the other hand, in the abscissa, the dimensionless distance from the beginning of the degradation is plotted. Thus, the degradation starts from 0 until reaching the $\tau_{95}$ at distance 1.

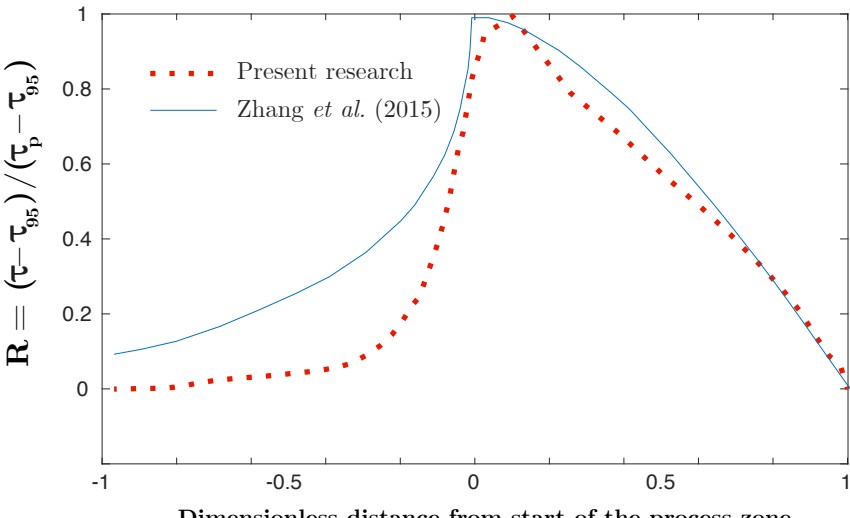

**Figure 6.** Comparison of the results of the dimensionless stress along the dimensionless distance between the proposed algorithm and the solution provided by Zhang et al. [54].

Results obtained in this research and the ones obtained by Zhang et al. [54] are not coincident, since they are different approaches; however, the overall trend, mainly after the beginning of the degradation, is similar to the reference research. This application allows us to assess the performance of a degraded layer of soft clay, which has been proved to perform similar to other validated studies.

### 4.2. Strip Footing Load

Following, the behavior of the soil under different loading rates of a strip footing is analyzed. Similarly, the viscous properties of the soil are varied in order to extend the analysis to the behavior of the soil. This classic problem has been extensively used to verify the solutions provided by numerical models under visco-plastic conditions. The two main features to assess are the mechanism of failure and the behavior of the reaction forces at different visco-plastic scenarios. These results have been previously presented in the work of Pastor and coworkers [37,60], where an incremental velocity downwards at the base of the strip footing is applied as the loading condition. In the proposed application, it is applied as a negative displacement according to the following equation:

$$u_y = u_f \left( t / t_f \right)^2,$$

where $u_f = 0.04$ m. and $t_f = 4$ s. The geometry and soil parameters can be seen in Figure 7. Parameters of the eigendegradation algorithm are also depicted. Opposite to the previous application, in this case, there is no weak layer: the whole domain acts as a visco-plastic degradable soil.

The first calculation is made by activating the degradation part. This degradation acts as a softening of the material following the proposed exponential expression. It is known how the softening of the material boosts the formation of shear bands. In Figure 8, the mechanism of failure is depicted. In [60], there is a study of the influence of the discretization size and the parameters of the meshfree model. Optimal options achieved in the aforementioned study have been activated here.

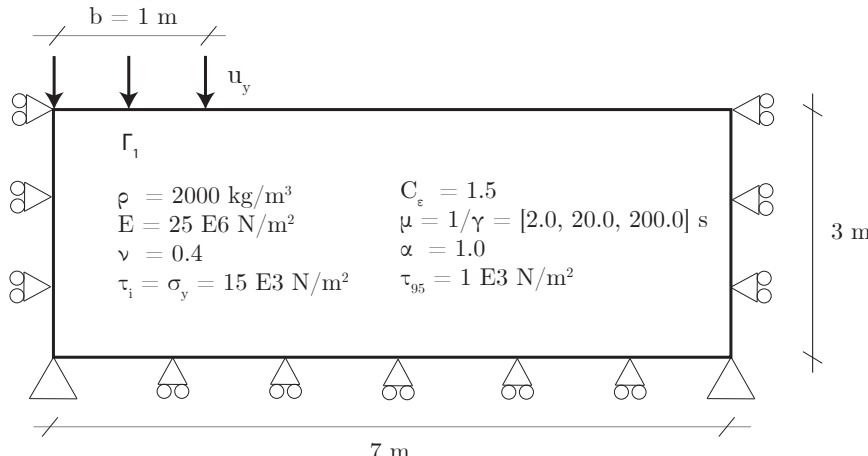

**Figure 7.** Geometry, material parameters and loading conditions of the strip footing problem.

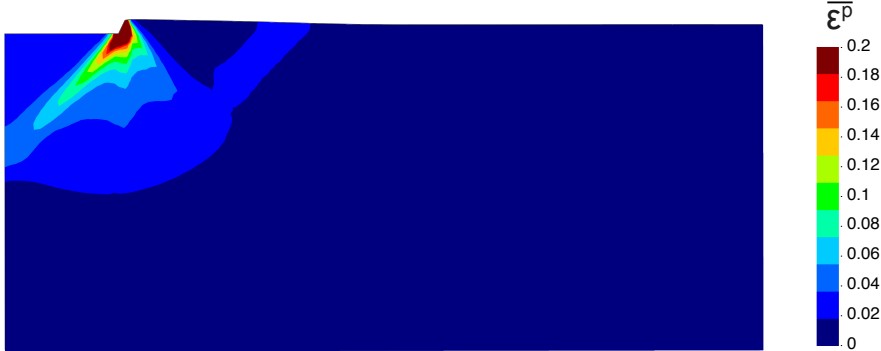

**Figure 8.** Equivalent plastic deformation of the footing problem using von Mises law with softening through degradation at the final of the simulation.

In order to verify the performance of the full model, different viscous parameters have been employed in the calculation of the failure load of the strip footing. Establishing the sensitivity parameter $\alpha$ as 1.0, the viscous parameter has been varied with different values (see Figure 7). The obtained results have been depicted in Figure 9 for different values of $\mu$. The smaller the value of the $\mu$ parameter (i.e., large values of $\gamma$), the higher the final loading is obtained from the footing loading, as expected from the viscous model. In addition, in the dashed line, the reference value obtained by Navas et al. [60] is depicted. This line can be considered as the value of a pseudo-static load is applied without any rigidization due to the viscous behavior. This value is very close to the one obtained with $\mu = 200$ s.

Similarly, depending on the loading rate, the material can stiffen and provide a bigger response of the reaction forces. Thus, for $\mu = 200$ s, three different loading rates have been tested. The obtained results have been depicted in Figure 10 for different values of $t_f$, in which this parameter is the final time of application of the imposed displacement $(u_y = u_f \left( t/t_f \right)^2)$. The quicker the application of the displacement, the higher the final loading is obtained from the footing loading, as expected from the viscous model. In addition, the dashed line depicts the reference pseudo-static value obtained by Navas et al. [60] also in this figure, which is close to the slowest case.

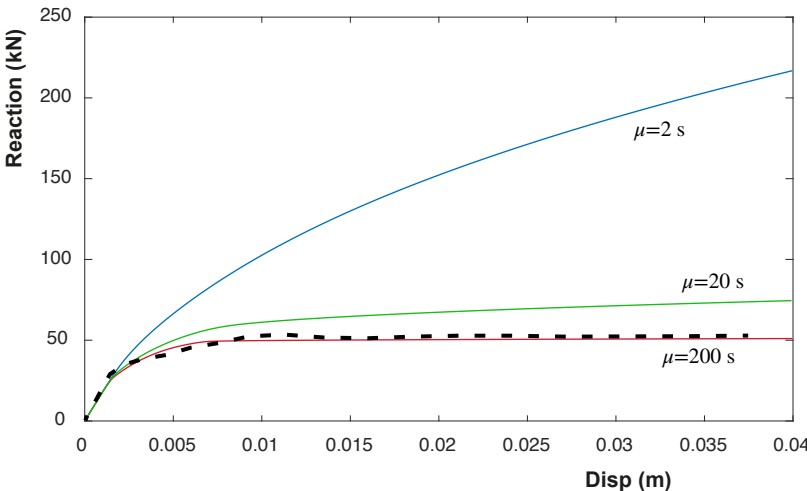

**Figure 9.** Obtained reaction with different $\mu$ values for the problem of the strip footing using visco-plastic von Mises law. The dashed line represents the pseudo-static behavior.

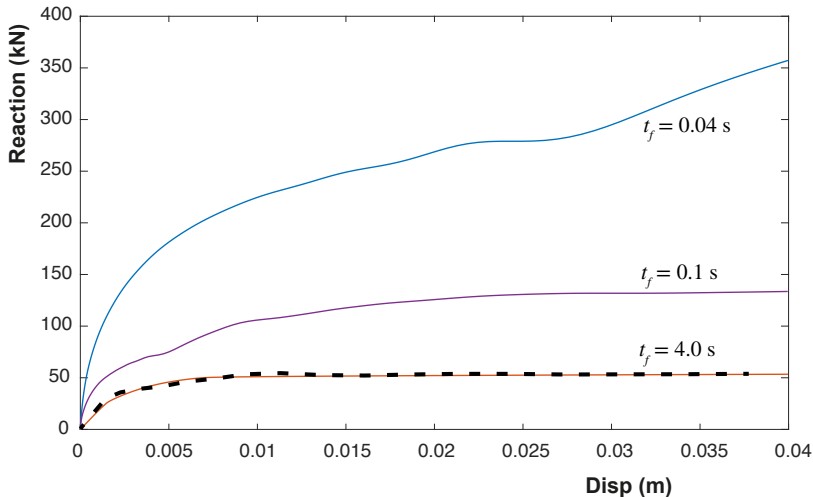

**Figure 10.** Obtained reaction with different loading rates for the problem of the strip footing using visco-plastic von Mises law, in which $t_f$ is the final time of the application of the load. The dashed line represents the pseudo-static behavior.

### 4.3. Vertical Cut

This final application allows understanding the potentiality of the proposed methodology. A weak layer is supposed in a soil with a vertical cut on the left side (Figure 11). This weak layer is located, forming a 45° angle. This layer, the thickness of which is 1 m, will be considered as plastic. Von Mises yield surface is employed, and its degradation is modeled through both *eigendegradation* and traditional softening in order to assess the performance of the former one compared with the latter. Out of the weak layer, the soil is considered elastic since its failure is far from the failure of the weak layer. Parameters of both models are presented in the right part of Figure 11. In the traditional softening model, no eigendegradation parameters are needed. Instead, a negative hardening of 200 kPa is employed. This parameter is not employed in the eigendegradation simulation.

The soil can be considered infinite on the right and on the bottom of the model; thus, any movement in these directions is prevented. The 12 first meters are modeled for the sake of simplicity.

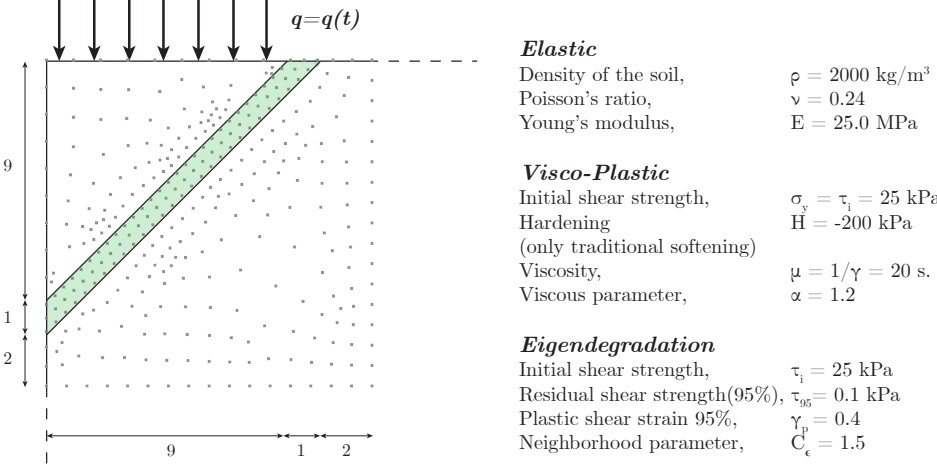

**Figure 11.** (**Left**:) Geometry of the vertical cut analyzed through *eigendegradation* and softening models and the location of the weak layer and the loaded zone (units in meters). (**Right**): Parameters of the employed models.

The top left part is loaded by a surface load as shown in Figure 11. This load is composed by two different waves as sketched in Figure 12.

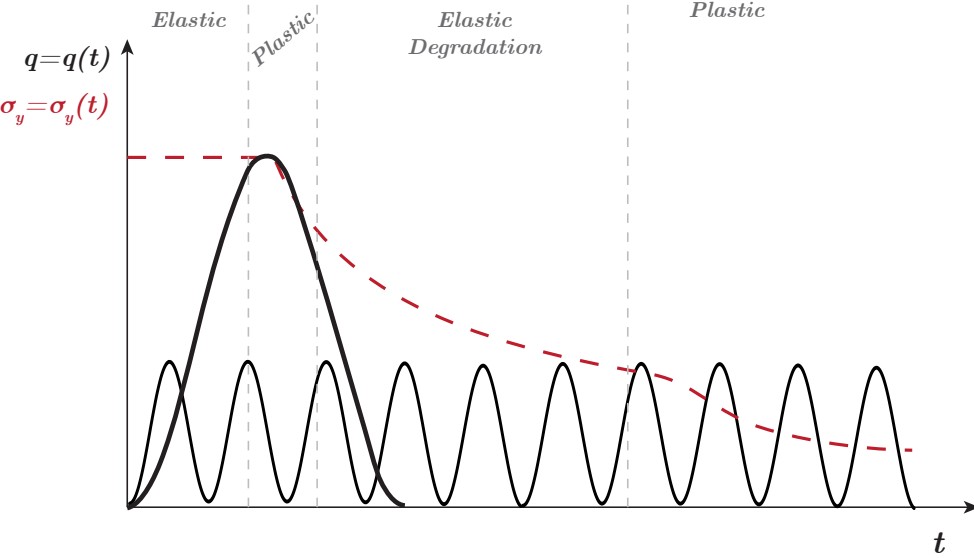

**Figure 12.** Scheme of the load and the yield strength along the time.

Both waves follow the expression:

$$q_\alpha(t) = A_\alpha \cdot [1 - cos(\omega_\alpha t)]$$

where $\alpha$ varies for each of both loads. The bigger load is the one that provokes the triggering of the plastic mechanism. It can be considered as an abnormal scenario that may lead to catastrophic consequences in the short or in the long time. Their parameters are $A_1 = 11$ kPa and $\omega_1 = 0.1\pi$ rad/s. This load is applied only in the first 20 s of simulation. The secondary load is of a lower magnitude. It could be considered as a usual load that the soil suffers permanently; it is not capable of provoking the breakage of the slope by itself. The amplitude of this load is half of the first one, $A_2 = 5.5$ kPa, and the frequency is much higher, $\omega_2 = 3\pi$ rad/s. This load is maintained through the 80 s of the simulation.

Following, in Figure 13, the evolution of both shear strength and the equivalent plastic strain are depicted along the time for both eigendegradation and softening models. Both OTM and FEM are carried out. The latter is made through the commercial software Abaqus. On the left, the shear strength is plotted. The first part of the figure is loaded with the first load: the material reaches the yield stress close to 10 s and starts to decrease the strength until 15 s. Both materials, until this point, behave similarly. Observing the right figure, we can see how both models obtain plastic strain until 15 s as well. Obviously, the amount of shear strain is different, since one law is logarithmic (eigendegradation) and the other one is linear. After this point, the eigendegradation model accumulates shear strain (elastic in this case) that makes the shear strength decrease. This elastic shear strain comes from the second law (the one with small amplitude). We know that it is elastic strain since, in the right figure, no accumulated plastic strain is obtained from 15 s to 50 s. From this point on, the equivalent plastic strain increases drastically. It is translated in an increment of the descent of the yield stress. However, since this accumulated strain provoked by the second load is elastic, no variation of the shear strength with the traditional softening model is observed. This pattern in obtained through both OTM and FEM simulations.

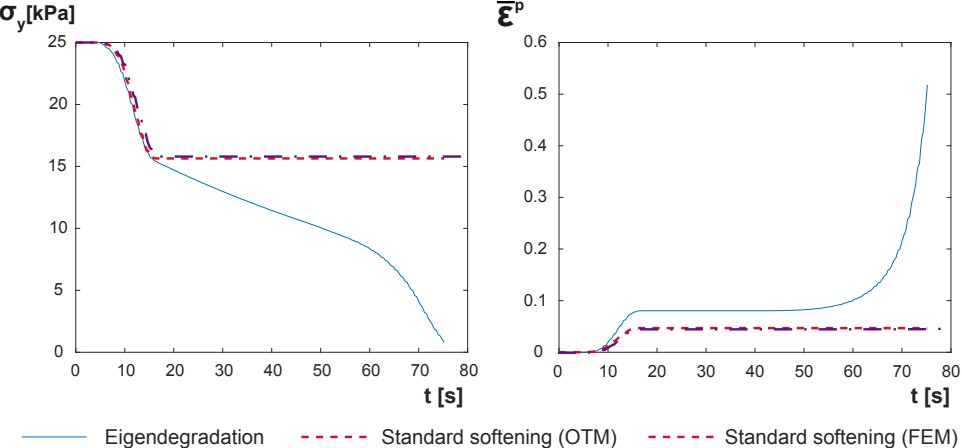

**Figure 13.** Evolution of the shear strength and the equivalent plastic strain along the time for both eigendegradation and softening models.

Another observation that arises from Figure 13 is the capability of the model to be sensitive to both loading and unloading conditions of the load, i.e., any variation of the strain, positive or negative, makes the material degrade and lose shear strength. It is seen in the slope of the yield stress curve from 15 to 50 s, which remains constant along the whole loading cycle and is equal in the loading or unloading branch.

Finally, in Figure 14, the distribution of the equivalent plastic strain in the deformed model at four different times is depicted. The chosen moments were: (i) the peak of the first load (around 15 s), (ii) the beginning of the secondary plastification of the material (50 s), (iii) moments before the failure of the slope (75 s) and (iv) moments after the triggering of the slope. Compared to the one obtained with the standard softening model, Figure 15, it can be seen how, after 20 s, the latter remains unalterable.

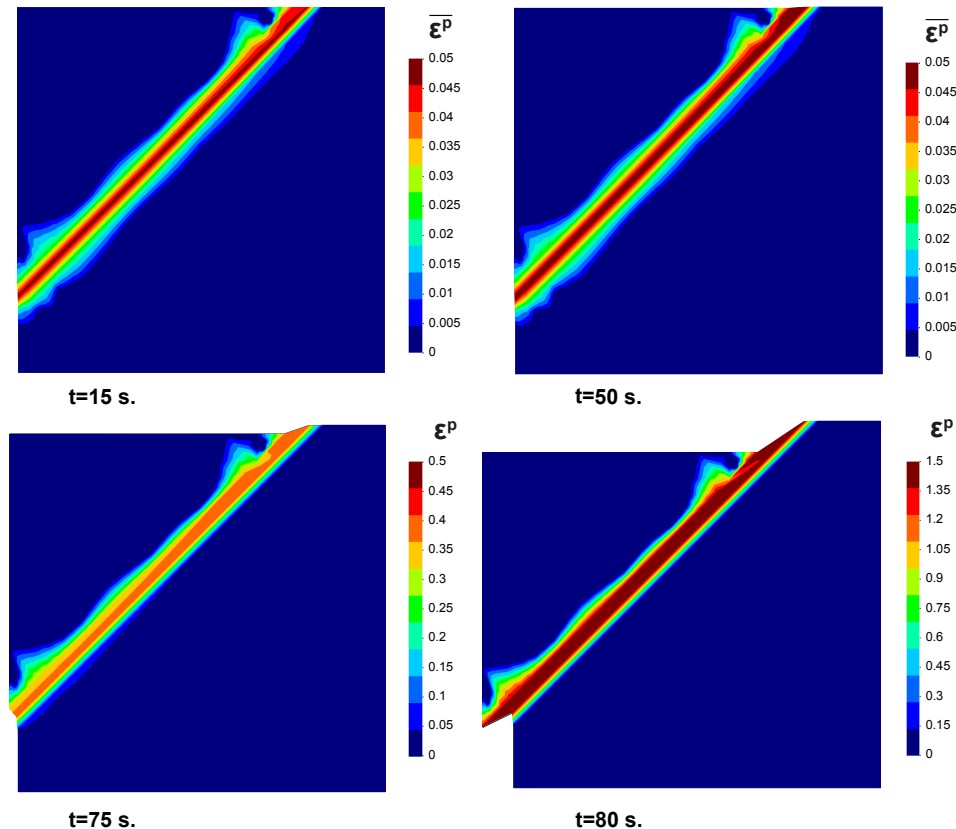

**Figure 14.** Distribution of the equivalent plastic strain in the deformed model obtained with the eigendegradation model at 4 different times.

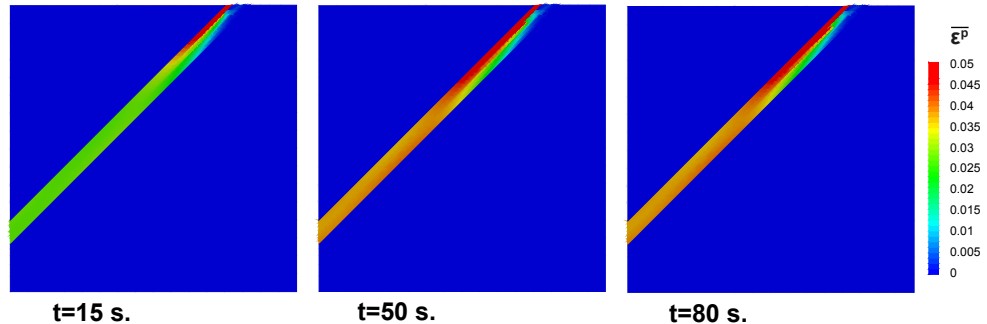

**Figure 15.** Distribution of the equivalent plastic strain in the deformed model obtained with the standard softening model (FEM) at 3 different times.

## 5. Conclusions

A new visco-plastic degradation model has been proposed in order to simulate the behavior of layers of soft clays. The main aim of this model is to eliminate the mesh dependence shown by other methodologies available in the literature. Taking benefits of the concept of eigendeformation, this model can reproduce both non-local and localized (shear bands) failures. The non-local performance of the proposed algorithm has been considered following concepts of non-local meshfree models capable of reproducing both brittle (eigenerosion) and quasi-brittle (eigensoftening) behaviors. Thus, the proposed algorithm takes the name of *eigendegradation*.

Two main properties define the proposed constitutive model: softening due to degradation and visco-plastic behavior. Thus, two different examples are provided in order to validate both properties.

The first example reproduces a shear test of a soil with a thin soft layer. This test shows the performance of the degradation behavior from the beginning of the reduction of the shear strength to the final of this reduction, obtaining a residual strength. This result has been compared to other similar work of degradation of clays, obtaining very close curves, which let us think the behavior of the proposed algorithm when modeling this material is correct.

The second example studies the behavior of a soil loaded by a strip footing. In this case, the visco-plastic concept is the one to be assessed. Since the von Mises yield surface is employed, the traditional Prandtl mechanism is obtained. Moreover, a hardening behavior is observed either when the viscosity becomes higher or when the load is applied quicker. This example allows us to validate the viscous behavior, as well as to verify that the degradation acts as a softening of the material that helps to define clearer the shear bands formed in the plastic mechanism.

Finally, a vertical cut of soil is modeled. This soil contains a 45° soft layer. A compound load is applied, the first part being the one that causes the beginning of the degradation of the material and the last part being the one that causes the whole degradation of the shear strength and the final failure of the slope. In comparison with a traditional softening law, the proposed one is capable of producing degradation with elastic accumulated strain, with the accumulation under both loading or unloading conditions. This behavior is seen in many failures of this type of soft layer, where a load is much lower than the critical one, but it is held in time, causing the fatigue of the material and the final failure of the soil structure.

The present algorithm has performed successfully with the proposed applications. The algorithm is robust, and its conjunction with any particle-based numerical technique (OTM in this research) presents an interesting feasibility. The proposed methodology is able to reproduce previous results with degradation models (example 1) and visco-plastic materials (example 2). Moreover, as it is observed in example 3, once the mechanism is activated, the model is capable of capturing the degradation of the material even in small elastic cycles, which allows reproducing the triggering of a landslide from a very low loading range.

Further research can be made in order to be able to reproduce a wide range of problems. First of all, this model should be calibrated against experimental tests such as the shear test. Moreover, only the von Mises yield surface has been validated in this manuscript. Although it is able to reproduce the undrained conditions of the soil, more sophisticated yield surfaces, such as the Cam–Clay one, would improve the type of problems to be modeled. The soil should also be modeled as a bi-phase material, including the water in the formulation of the problem. Finally, some other spatial discretization, such as FEM or MPM, could be employed to assess the performance of this algorithm.

**Author Contributions:** Conceptualization, S.L.-Q. and D.M.; software and validation, P.N.; resources, Á.Y.; supervision, M.M.S. All authors have read and agreed to the published version of the manuscript.

**Funding:** This research was funded by the Ministerio de Ciencia e Innovación, under Grant Number, PID2019-105630GB-I00; and the European Research Council-H2020 MSCA-RISE, Grant Agreement No 101007851 (DISCO2-STORE), being both greatly appreciated.

**Institutional Review Board Statement:** Not applicable.

**Informed Consent Statement:** Not applicable.

**Data Availability Statement:** Contact to pedro.navas@upm.es.

**Acknowledgments:** The administrative and technical support of both University College London and Universidad Politécnica de Madrid is greatly appreciated.

**Conflicts of Interest:** The authors declare no conflict of interest.

## Abbreviations

The following abbreviations are used in this manuscript:

FEM      Finite Element Method
OTM     Optimal Transportation Meshfree
SPH      Smooth Particle Hydrodynamics
MPM    Material Point Method
ALE      Arbitrary Eulerian–Lagrangian

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
