# Peer review of "Eigendegradation Algorithm Applied to Visco-Plastic Weak Layers"

_applsci, doi:10.3390/app12168175_

Round 1
Reviewer 1 Report
Please see the attachment.

Author Response
Thanks for the review and for the good comments on the paper. We would like to thank you also the carefully reading and the finding of all the typos. Here we reply to all your comments:
Main comments:
1. It is suggested to present some comparison results to show the advantages of the proposed methods. These comparison results can be obtained from some existing works or some commercial software.
A calculation through ABAQUS has been added, in the third example, with the visco-plastic model. figure 13 is modified and a new figure, 15, appears. Obviously, it was not possible to add any eigendegradation result since it is not available in any commercial software.
2. This paper interpolates with Optimal Transportation Meshfree, a mesh-free method that has been shown to perform well in geotechnical problems. What is the advantage of this method over finite element? Please give some discussion.
A discussion has been added (in blue) in the introduction of the document.
Minor Comments
Lines 91, 95, 100, 105, 167, 178, 215 and 306: No indentation required.
Fixed
Cepsilon line 242 should be rewritten as Cepsilon
FIxed
In Figure 4, 2ℎC should be changed to 2ℎCepsilon
Fixed
There is an extra "+" sign in equation (31).
Fixed
Reviewer 2 Report
This paper is well written. I presume the detailed formulations are correctly derived as I can see that the analytical outputs are reasonable from my understanding. I thus do not have questionings to this paper.
The conclusion part should be revised, as many of the them are descriptive only. Please follow a conclusion writing style.
In your briefing of meshfree techniques for geotechnical study, a recent SPH simulation work is worthy of discussion in lines 26 to 27. For details, please refer to https://doi.org/10.1016/j.gsf.2020.02.003
Author Response
Thanks for the review and for the good comments of the paper. Here we reply all your comments:
The conclusion part should be revised, as many of them are descriptive only. Please follow a conclusion writing style.
Revised in blue color in the corrected manuscript.
In your briefing of meshfree techniques for geotechnical study, a recent SPH simulation work is worthy of discussion in lines 26 to 27. For details, please refer to https://doi.org/10.1016/j.gsf.2020.02.003
Added
Reviewer 3 Report
Dear Editors and Authors,
The article is a good study of the behavior of soils with the consideration of the time factor and the failure of weak layers. A new calculation algorithm based on damage mechanics and bond shear with time progression was proposed. The algorithm was applied to 3 specific cases, achieving good calculation agreement.
The article is eligible for publication in Appl. Sci.
Minor corrections are indicated below:
Line 112: In some lines in "etal." there is no space between "et" and "al.", and in line 233 there is a space between "et" and "al.". Please standardize the notation throughout the article.
Figure 4: Shouldn't the "P" be italicized?
Line 206-209: there are " empty squares" instead of symbols.
Line 242: is the notation of the symbol "Cepsilon" correct?
Figure 10: Please add quantities symbols on the vertical and horizontal axes.
Figure 11: Space between "Left:" and "Geometry."
Figure 12. Please consider adding units on the vertical and horizontal axes. Please correct the crossing, with the last cycle of the curve below the t-axis (1 mm of the curve), which means that q<0 (to the left of the "t" symbol).
Figure 13: Graph on the right. Is the marking of the symbol epsilon_p correct (canopy) compared to the notation of formula 4.5. on page 9/21?
With regards.
Author Response
Thanks for the review and for the good comments of the paper. We would like to thank you also the carefully reading and the finding of all the typos. Here we reply all your comments:
Line 112: In some lines in "etal." there is no space between "et" and "al.", and in line 233 there is a space between "et" and "al.". Please standardize the notation throughout the article.
Fix to "et al" in the whole document.
Figure 4: Shouldn't the "P" be italicized?
Fixed
Line 206-209: there are " empty squares" instead of symbols.
It is correct. It means any quantity with dots means time derivative of it.
Line 242: is the notation of the symbol "Cepsilon" correct?
Fixed
Figure 10: Please add quantities symbols on the vertical and horizontal axes.
Do not understand this comment, sorry.
Figure 11: Space between "Left." and "Geometry."
Fixed
Figure 12. Please consider adding units on the vertical and horizontal axes.
It is just a scheme, without units.
Please correct the crossing, with the last cycle of the curve below the t-axis (1 mm of the curve), which means that q<0 (to the left of the "'" symbol).
Fixed
Figure 13: Graph on the right. Is the marking of the symbol epsilon p correct (canopy) compared to the notation of formula 4.5. on page 9/21?
Fixed
Round 2
Reviewer 1 Report
The authors have revised the manuscript according to my suggestions. I am happy to recommend its publication.